# Sensory Preference and Professional Profile Affinity Definition of Endangered Native Breed Eggs Compared to Commercial Laying Lineages’ Eggs

**DOI:** 10.3390/ani9110920

**Published:** 2019-11-05

**Authors:** Antonio González Ariza, Ander Arando Arbulu, Francisco Javier Navas González, Francisco de Asís Ruíz Morales, José Manuel León Jurado, Cecilio José Barba Capote, María Esperanza Camacho Vallejo

**Affiliations:** 1Department of Genetics, Faculty of Veterinary Sciences, University of Córdoba, 14071 Córdoba, Spain; angoarvet@outlook.es (A.G.A.); anderarando@hotmail.com (A.A.A.); 2Instituto de Investigación y Formación Agraria y Pesquera (IFAPA), Camino de Purchil, 18004 Granada, Spain; franciscoa.ruiz@juntadeandalucia.es; 3Centro Agropecuario Provincial de Córdoba, Diputación Provincial de Córdoba, 14071 Córdoba, Spain; jomalejur@yahoo.es; 4Department of Animal Production, Faculty of Veterinary Sciences, University of Córdoba, 14071 Córdoba, Spain; cjbarba@uco.es; 5Instituto de Investigación y Formación Agraria y Pesquera (IFAPA), Alameda del Obispo, 14004 Córdoba, Spain; mariae.camacho@juntadeandalucia.es

**Keywords:** professional profile, sensory attributes, academic level, product knowledge, production context awareness, cuisine applicability

## Abstract

**Simple Summary:**

A local breed’s particularities may provide eggs with sensory properties which may overcome laying lineage, regardless of their production system characteristics. Hence, methods clarifying what the appreciation of a certain product is like can outline the actions required to improve the market value of that product. Affine and non-affine profiles were defined based on the information provided by sixty-four professionally-instructed panelists on sensory attributes, diet habits, production context awareness, product consciousness, cuisine applicability and panelist attributes. Egg consumption was lower in non-affine profile professionals, as were the scores provided to sensory attributes. The higher the knowledge about Utrerana breed, the greater the importance provided to the ecological and autochthonous nature of the products. The level of study, gender and age are crucial factors to consider when approaching the commercialization of Utrerana hen eggs. Conclusively, defining consumer profiles among professionals of the cuisine sector may improve the profitability of Utrerana eggs and may help educating non-affine profiles, something key to the success in product appreciation.

**Abstract:**

This study aimed to compare Utrerana native hen eggs’ sensory properties to Leghorn Lohmann LSL-Classic lineage’s commercial and ecological eggs through free-choice profiling. Second, affine and non-affine profiles were defined using the information provided by professionally-instructed panelists on six sets (sensory attributes, diet habits, production context awareness, product consciousness, cuisine applicability and panelist attributes) using nonlinear canonical correlation analysis. Sixty-four instructed professional panelists rated 96 eggs on 39 variables comprising the above-mentioned sets. Observers reported a significantly higher appreciation (*p* > 0.05) towards yolk color, odor, flavor, texture, overall score, and whole and on plate broken egg visual value when Utrerana eggs were compared to the rest of categories. Professional Profile A (PPA), or egg non-affine profile, consumed less eggs and provided lower scores to sensory attributes than Professional Profile B (PPB), or affine profile. Additionally, PPB accounted for higher knowledge about the Utrerana breed and provided greater importance to a product’s ecological and autochthonous nature. PPA was generally characterized by women under 20 years old with no higher studies, while PPB comprised 21–40 years old men with secondary studies. In conclusion, defining professional profiles enables correctly approaching market needs to improve the profitability of Utrerana eggs, meeting professional demands and educating non-affine profiles.

## 1. Introduction

The Utrerana hen is a rustic medium-sized Spanish endangered breed with four feather varieties (partridge, Franciscan, black, and white). Initially regarded as a white egg laying hen (120–180 eggs/year, mean ~64 g/egg for the entire laying period) raised on family farms, a traditional meat/egg aptitude has been addressed in literature and is reappearing in the market scene [1].

In 2017, around 92% of eggs were produced by laying hens. The annual growth rate of egg production was approximately 0.6 million tons per year from 2000 to 2016, with a total of 1416 trillion tons of eggs, equivalent to 80 million metric tons [2]. However, consumer tastes constantly evolve towards the obtainment of quality food products with special properties, aimed at a specialized market and obtained through sustainable production systems [3]. In this context, Utrerana eggs have shown to have differentiated internal and external characteristics from commercial line eggs [4]. Hence, the need for animal genetic resources proper conservation has greatly increased, since these resources present economic, scientific and cultural interests [5].

Modern livestock production high specialization increasingly threatens animal genetic resources diversity [5]. For instance, commercial herds are based on exploiting few genetically selected breeds for intensive production. However, specialized breeds do not guarantee a genetic reserve for the future. After long periods of natural selection and evolution, variability loss compromises diversity and characteristics such as adverse conditions adaptation, which conforms an invaluable protein rich source [6].

Although nutrition has been suggested to be more influential than genetics in egg chemical composition, Washburn [7] found evidence for a heredity component. Egg quality is directly related to characteristics determining consumer acceptability. For instance, some traits, such as egg weight and dense white height, are highly valued by consumers [8]. However, there are a number of egg sensory attributes that are more difficult to assess, for which taster panels are used.

Food chemical composition is appreciated through the taste sense. Still, some caution should be taken to transform taste sensations into reliable measures, given the existence of interpanelist variability sources that cannot be eliminated even after training [9].

Panelists have been suggested to be unable to routinely describe the sensory attributes they perceive; hence, profiling methods must be homogenized (same sensory lexicon, among others). To prevent this, techniques such as hedonic scale measurements are used [10].

Developing sensory tools to define potential consumers profiles and attitudes towards products has always concerned food scientists. Tests are diverse and range from those analyzing the process of standardization for food evaluation and perception derived from product/consumer interaction to panelists sensation elaboration and verbalization [11].

Empirically determining panelists discriminating capacity for organoleptic characteristics commonly involves the implementation of linear or logistic regression models used in preference surveys and social epidemiology [12]. However, in the case of multivariate analysis, non-linear canonical correlation analysis more appropriately allows to map a series of explanatory factors in correlation to different sensory attributes and eggs’ cuisine applicability [13].

First, we aimed to determine the ability of panelists to discriminate organoleptic characteristics across egg type categories basing on hedonic scales. Second, we inferred different professional profiles regarding their affinity to eggs and their personal context, as a strategy to plan potential marketing strategies to reinforce affine professionals and to attract non-affine ones, to promote autochthonous breed conservation strategies relying on the improvement of their profitability.

## 2. Materials and Methods

### 2.1. Free-Choice Profiling (FCP)

#### 2.1.1. Professional Panel Description

Sixty-four professional background instructed panelists (students and teachers of the Hospitality School of Córdoba and Granada), with ages ranging between less than 20 to over 50 years old and a minimum formation of 30 h/week during a two-module professional expertise course (Royal Decree Law 687/2010), were recruited to participate in a short-term sensory evaluation study. The number of panelists included in the hedonic test complies with recommendations for best practice in sensory and consumer science proposed by Hough et al. [14] for primary and processed food.

The recruitment was performed after a filter questionnaire, which included demographic and socio-economic information (gender, age and academic level), food products consumption frequency and egg cuisine applicability willingness. No panelist was removed given the lack of missing values.

#### 2.1.2. Sampling

Ecological and commercial eggs (A category and of the M class (53–63 g) belonged to white shell Leghorn Lohmann LSL-Classic lineage. Utrerana and Leghorn Lohmann LSL-Classic lineage commercial hens were managed in individual cages (50 × 62 × 41 cm) following Council Directive 1999/74/EC of 19 July 1999, setting the minimum standards for the protection of laying hens at the Centro Agropecuario Provincial de Córdoba (Spain). Feed and water were available ad libitum. All birds were reared according to the regulations of the European Union (2010/63/EU) in their transposition to the Spanish law (RD 53/2013). Stocking density was four animals per each m^2^, nest box density consisted of 29 animals per m^2^. Circle waterers of 5 cm diameter and 41 cm feeder allotment/space were provided for each animal. Wood shavings were used as a floor substrate covering the floor to a depth of approximately 1 cm. Nest box substrate consisted of plastic turf mats covering the floor at a depth of approximately 1 cm. Further information regarding maintenance system of Leghorn Lohmann LSL-Classic lineage commercial hens and Utrerana native breed can be found in González Ariza et al. [4].

Ecologic/organic eggs were obtained from Leghorn Lohmann LSL-Classic lineage hens. The birds were placed in pens comprising a ceiling covered surface of 41.6 m^2^ and a free-roaming surface of 1000 m^2^ following the Council Regulation (EC) No 834/2007 on organic production and labelling of organic products with regard to organic production, labelling and control. A total of 20 cm of perch was provided per bird. Food and water were available ad libitum. All birds were reared according to the regulations of the European Union (2010/63/EU) in their transposition to the Spanish law (RD 53/2013). Stocking and nest box density, waterers and feeder space followed the requirements stated at the regulation referenced above. Wood shavings were used as a floor substrate covering the floor at a depth of approximately 1 cm and nest box substrate consisted of plastic turf mats covering the floor at a depth of approximately 1 cm as well.

A description of the recipe and chemical composition of the compound feed used for hen feeding in this study is provided in Appendix A. To avoid an effect of storage time on the sensory properties, evaluation of the samples was performed in two locations (Córdoba and Granada) simultaneously. A 10 min cooking time was determined after a preliminary boiling test performed on a random control group of 27 eggs to determine the duration (9, 10 and 11 min) that prevents overcooking. Eggs were first strained in cold water to prevent shelling during cooking. Sixty-four testing stations were set. Eggs were cut following their longitudinal axis. Each panelist tested and scored half an egg from Utrerana, half an egg from a commercial intensive production Leghorn Lohmann LSL-Classic lineage, and half an egg of the same lineage raised under ecological free-range conditions.

#### 2.1.3. Evaluation Sessions

Participants were received in a conference room and placed at individual blind stations under white lighting (700 lx ± 150 lx), as suggested by Guàrdia et al. [15]. First, panelists were briefed on the methodology and the procedure to allow for acquaintance with the vocabulary to describe the three egg types. Each sample was labeled with random three-digit code matched with the panelist number plus an additional random code to identify samples of the same egg type (Commercial eggs—386; Free-range eggs—745; and Utrerana eggs—639), in a randomized complete-block design. A maximum of three samples were presented to each panelist and were assessed in the same tasting session balancing the first-order and the carry-over effects [16], as suggested by Guàrdia et al. [15].

Eggs were evaluated at room temperature (20 °C) and presented on white ceramic plates covered with a food grade PVC film (oxygen permeability; 20,000 cm^3^/m^2^/24 h; water-vapor transmission rate 2000 g/m^2^/24 h; Macopal, S.L., Lliçà de Vall, Spain) to prevent drying. Mineral water and 15 g golden delicious variety apple slices [17] at room temperature were provided for mouth rinsing and sense saturation reduction in participants between samples.

#### 2.1.4. Sensory Evaluation and Panelist Contextual Records

Each half egg was rated on six egg sensory attributes (yolk color, white color, odor, flavor, texture and overall score). The visual value of the whole egg and a broken egg on plate visual value were also scored for each egg type separately. Collaterally, panelists provided information on five additional sets comprising a total of 31 variables. These questionnaires later allowed characterization of the overall profile of the panel and the panelists’ preferences for the products tested. The panel comprised fairly equal amounts of females (44%) and males (56%).

Panelist context is defined by five sets of variables as follows: Panelist diet habits, production context awareness, product consciousness, cuisine applicability and panelist characterization. The definition for each set and its comprising variables and scales is shown in Appendix A. The scales used follow one unit increases to indicate panelists’ ratings. Egg sensory attributes were rated on a 1 to 8 hedonic structured or categorized scoring scale extracted from Anzaldúa Morales [18], except for white color, where panelists only provided answers for seven categories. The sensory attributes set was evaluated using an structured 100 mm line scale anchored with the following ordinal categories: (1) I extremely dislike it, (2) I dislike it a lot, (3) I dislike it moderately, (4) I slightly dislike it, (5) I like it, (6) I slightly like it, (7) I like it moderately, (8) I like it a lot and (9) I extremely like it, adapting the criteria in Anzaldúa Morales [18].

### 2.2. Free-Choice Profiling Interobserver Correlation Coefficient (ICC)

The intraclass correlation coefficient (ICC), based on multiple paired Cohen’s κ tests, was calculated to determine if there was agreement between the sixty-four panelists. Fleiss and Cohen [19] established repeatability guidelines for ICC interpretation as less than 0.4 (low), from 0.4 to 0.59 (reasonable), from 0.6 to 0.74 (good), and from 0.75 to 1.0 (excellent). As we used a random sample of consistent raters for all ratees, we used a “Two-Way Random” model. Then, 95% confidence intervals were computed. The ICC and 95% CI were calculated with the reliability analysis routine of the scale procedure of SPSS Statistics for Windows, Version 24.0, IBM Corp. (2016) (Table 1).

### 2.3. Scale Reliability

Scale internal consistency was studied using Cronbach’s alpha. As a general criterion, George and Mallery [20] suggest the following recommendations for evaluating Cronbach’s alpha coefficients: >0.9 is excellent, >0.8 is good, >0.7 is acceptable, >0.6 is questionable, >0.5 is poor and <0.5 is unacceptable. Variables with values over 0.5 were retained as they were able to explain the highest percentage of variance.

### 2.4. Quantitative Descriptive Analysis (QDA)

Variables and scales use agreement was performed at a preliminary open discussion involving 32 professional panelists, following the premises described in Anzaldúa Morales [18]. The same author reported the references used to illustrate the criteria for the variables on each set. Several training and refresher training sessions were set up to develop the different sensory attributes and normalize the panelists according to common perceptions [18]. Descriptors varied from 8 to 10 for each panelist.

### 2.5. Egg Type Sensory Attributes Difference Analysis

Descriptive statistics for the variables on each set are reported in Appendix A. Variables were not transformed and sorted into six sets considering their common nature, namely, egg sensory attributes, panelist diet habits, production context awareness, product consciousness, cuisine applicability, and panelist characterization. Shapiro–Francia tests were carried out with the .sfrancia routine of StataCorp Stata version 14.2. (Appendix A). As normality was not found, a Kruskal–Wallis H test was performed to study differences across variables. Afterwards, interlevel distribution and median differences among Kruskal–Wallis H significant variables were tested using the pairwise comparisons Dunn’s test and sorting medians respectively. If we test for multiple comparisons, the likelihood of incorrectly rejecting statistically significant differences between two or more levels (Type I errors) increases. The Bonferroni correction was performed to compensate for that increase. All nonparametric tests were carried out using the independent samples package from the non-parametrical task of SPSS Statistics for Windows, Version 24.0, IBM Corp. (2016) and results are provided in Appendix A.

### 2.6. Statistical Justification

Although some authors [21] have suggested Procrustes analysis to be one of the most common and strict techniques to analyze sensory attributes related to other aspects such as free choice profiling, it is only applicable when all variables measurement dimensions (p) have similar scales. Contrarily, this analysis renders inaccurate [22] if we do not only have different scales but also different measurement units.

The same authors suggest alternatives such as the nonlinear version of canonical correlation analysis, report results with a virtually perfect fit, which may be partly attributed to the freedom to choose non-linear transformations, which enables scoring traits on very different scales.

### 2.7. Non-Linear Canonical Correlation between Sets

A nonlinear canonical correlation analysis (OVERALS) was performed to determine interset similarities to maximize the variance in the relationships among two sets of numerical variables in a low dimensional space. Optimal scaling approach in OVERALS expands the standard canonical analysis as first, it allows more than two sets of variables, accommodating varying scaling standards [13]. Second, variables can be scaled as nominal, ordinal, or numerical in an intervariable integrative analysis of non-linear relations. Finally, instead of maximizing interset correlations, these are compared to an unknown compromise set that is defined by the object scores. OVERALS uses the “alternating least squares (ALS) algorithm”, to calculate the “fit function” and the “loss function”. The loss function states the difference between the number of chosen dimensions to the best calculated adaptation and shows the lack of fit of a solution, being within a p-dimensional case, the minimum equal to 0 and maximum equal to p. Loss represents the proportion of variation in object scores for each dimension and set in Table 2. The mean of sets is the average loss in sets and gives us the difference between the maximum and actual fits. Summation of average loss and fit is equal to the number of dimensions. Therefore, small loss values indicate large multiple correlations between weighted sums of optimally scaled variables and dimensions [23]. The eigenvalue can be calculated by dividing loss per dimensions, and carrying out 1 minus loss per dimension. The eigenvalue is a goodness of fit measure, which ranges from 0 to 1, indicating the level of relationship shown by each dimension, and the sum of these values is called total fit (Table 2), that is, the statistical index widely used in OVERALS to decide analysis solution dimensionality.

For visual mapping of the constructed space, we used the nonlinear canonical correlation analysis with the described six sets and their variables. Component loadings are the correlations between object scores and optimal scaled variables and are sorted in dimensions 1 and 2. These loadings act as coordinates of the variable points on the graph given below in Figure 1, Figure 2, Figure 3, Figure 4, Figure 5 and Figure 6 and help with illustrating the distribution of variables in a bi-dimensional space. To this aim, quantifications of multiple categories or numerical ranges are used. These quantifications present the center for all respondents belonging to one category and account for the importance of other variables from the set. Variables close to others have more similarities among interviewed persons than variables that are far apart. To interpret the dimensions obtained, attributes with loadings of over 0.5 [24] were the most effective variables in relationships among variable sets because they were positioned far from the origin (denoting the mean) [25] (Appendix A). The plots of centroids were labeled according to the categories in the scale for each variable and are presented in Figure 1, Figure 2, Figure 3, Figure 4, Figure 5 and Figure 6, showing how well variables separate groups of objects. Centroids were in the center of gravity of the objects. Matching clusters of categories in centroid plots need to be identified and interpreted to understand intervariable relationships [26].

Contrary to what happens in principal component analysis (PCA), for which a dimensionality criterion of explained variance over 80%, is required. When OVERALS is linked to FCP, if all variables are specified as ordinal, single nominal, or numerical, the maximum number of dimensions is the lesser of the following two values: The number of observations (n = 192) minus 1, or the total number of variables [26]. Then, we reduce the dimensions until we reach the maximum number of dimensions that explains the greatest percentage of variance at an acceptable loss level (Table 3). Single variables are only important when containing information independent from information of other variables of the same set [27].

In total, 39 variables with either a nominal, numeric or ordinal scaling level are included in the analysis (Appendix A). Variables can be classified into two or more sets and scaled as multiple nominal, single nominal, ordinal, or numerical and the interpretation of their direction is obtained from the position of projected centroids. Most of the variables considered in the present study are ordinal. This implies that the order of the categories within each variable must be preserved. Then, if actual and projected centroids are not separated, ordinal variables should have been considered as nominal [28]. As suggested by van der Burg and Dijksterhuis [29], sensory scores were reorganized into fewer new categories to minimize the existence of empty categories, we decided to adopt this organization system, thus minimizing the occurrence of unique marginal frequencies.

## 3. Results

Kruskal–Wallis H reported significant differences (*p* < 0.05) for all egg sensory attributes across egg type categories except for white color (*p* > 0.05). Dunn’s tests reported egg categories were significantly different, with Utrerana egg reporting the highest median (5), followed by ecologic (3), and commercial (2). The Utrerana egg scored one median-unit higher than commercial lineage, which also presented a significantly lower score for flavor, overall and on plate broken egg visual value when compared to ecologic eggs (1 point higher) and Utrerana eggs (2 points higher), with no significant difference between Utrerana and ecologic eggs. Texture was only significantly different between commercial and ecologic eggs (*p* < 0.05), with the latter reporting a one-point-higher median than commercial or Utrerana eggs. Commercial eggs’ whole egg visual value was significantly different to that of ecologic and Utrerana eggs.

Single ICC, determining how a single observation taken at random may correlate to another single observation, was 0.105, 0.120 and 0.125, for commercial, Utrerana and ecologic eggs, respectively. This could be expected, given that we were considering panelists’ personal appreciation of certain products, and no correlation should be expected beforehand as they may be strongly conditioned by subjective factors. However, average ICC and Cronbach’s alpha, that is, how consistent the whole panel of panelists is on average, were 0.800, 0.826 and 0.829, for commercial, Utrerana and ecologic eggs, respectively, reporting an excellent repeatability. This suggested the survey and scales used were sound and the panel was properly instructed and reliable.

Eigenvalues were high (0.673 and 0.645 for dimensions 1 and 2, respectively). Hence, the actual fit value was 1.318. A bi-dimensional solution was chosen, so 1.318/2 = 65.9% of the variation was calculated in the analysis, with 0.673/1.318 51.1% of the actual fit calculated by the first dimension and 0.645/1.318 48.9% by the second dimension.

Table 2 shows a summary of loss functions for each dimension and set. Average loss was 2 − 1.318 = 0.682 in our study and not necessarily high. The number of dimensions was equal to 2 (0.682 + 1.318). The single and multiple fit of variables is presented in Table 3. Component loadings are presented in Appendix A. The visual maps depicted in Figure 1, Figure 2, Figure 3, Figure 4, Figure 5 and Figure 6 are defined by all variables listed in Appendix A. Those variables, which in sum showed a multiple fit of more than 0.1 (Table 3), may play a more important role in the explanation of variance. Variable values, which are not displayed, were mainly spread around the axis of coordinates. By neglecting this proportion of data none of the influential values are lost, but the readability of figures is improved. Component loadings (interpanelist agreement) are shown for each variable separately in Appendix A. Dimension 1 shows that the panelists agree very much on the attribute whole egg visual, fish consumption, drug prohibition, commercial egg price, free range egg price, ecological egg price, ecological product status relevance, and salad applicability. The second dimension shows panelist agreement is dominated by environmental aspects, egg applicability in desserts, and academic level.

Apparently, when analyzing each set separately, yolk color, flavor and overall score were the attributes on which the panelists agreed less. These lowest agreement values are also reported for vegetable and meat consumption, hen welfare and genetically modified organism (GMO) banning, Utrerana knowledge, seasonal product conception or egg applicability in soup.

Two very distinct professional profiles are identified regarding attitudes towards eggs (Figure 1, Figure 2, Figure 3, Figure 4, Figure 5 and Figure 6), Professional Profile A—non-affine profile (PPA, in red in plots), and Professional Profile B—affine profile (PPB, in green in plots). The egg sensory attributes set is assessed in Figure 1. PPA scored yolk color from 2 to 4. The most of the observers from PPB scored yolk color higher than 5 out of 8 levels in the scale. PPA normally scored white color 1 or 2, while PPB scored white yolk from 3 to 6. PPA scored odor from 1 to 4, while PPB scored odor from 4 to 8. PPA scored flavor from 2 to 4, while PPB scored it from 5 to 8. PPA scored texture with the values of 2, from 4 to 5 and from 7 to 8, while PPB provided constantly increasing values from 1 to 7. For the overall score, PPA provided scores of 2 or 8 while PPB scores increasingly ranged from 4 to 7. For whole egg, PPA provided 4 or 8 scores, while PPB provided values from 2 to 6 (excluding 4). The on-plate broken egg value was irregularly scored by PPA and PPB.

PPA does not usually consume eggs, while PPB consumes more than four eggs a week more than two days per week (Figure 2). PPA either does not consume vegetables or consumes them from six to seven days a week, while PPB consumes vegetables from one to five days a week. PPA consumes fruit six to seven days a week while PPB or does not eat fruit or eat it less than five days a week. PPA eats meat one day per week while PPB eats meat more than two days per week. Fish and dairy consumption habits were reported by PPA in six to seven days, while PPB used to consume fish and dairy products less than five days a week or did not consume them at all. PPA did not consume ecological products while PPB did consume ecological products.

PPA provided a low importance to hen welfare, while PPB provided it with a higher importance (Figure 3). Contrastingly, PPA provided the highest importance for hens being kept in free range conditions in the scale, while PPB scored differently from 1 to 7 and from 8 to 9. PPA scored drug prohibition importance with 1 to 2 values and 4, while PPB scored its importance from 4 to 6 and from 8 to 9. A more irregular trend, supported by the low component loadings (Appendix A), was described in both profiles for environmental respect and GMO banning.

PPA scored product closeness from 1 to 3, while PPB scored it from 4 to 8. PPB was acquainted with the Utrerana breed while PPA was not (Figure 4). Contrary to PPB, PPA progressively misattributed the highest prices to commercial, free range and ecological eggs, respectively, and provided the lowest importance to ecological products or to the product being linked to an Andalusian autochthonous or endangered breed. The importance conferred to the seasonal attribute of the product described an irregular scoring pattern for both PPA and PPB.

PPA presented a lower trend to use eggs in desserts, as appetizers, in pasta, soup or salad than PPB (Figure 5). PPA mostly comprise women under 20 years old with no studies, while PPB comprised men from 21 to 40 years old and with secondary studies (Figure 6).

## 4. Discussion

The eigenvalues of the two dimensions that result from the nonlinear canonical correlation analysis are quite high, with 0.673 for the first dimension and 0.645 for the second dimension. Our total fitness value of 1.318 can be considered appropriate for this type of treatment [29], as it has been reported by several authors and food products. Other two-dimensional solutions reported in the literature have produced total fit indexes of 1.644 in apples, 1.763 in luncheon meat, 1.192 in water and 1.856 in cheese [24,29]. This makes the conclusions driven from the present study valid and reliable.

European consumers prefer darker yolks, given the psychological healthier egg qualities misattribution. Observers scored Utrerana yolk color and odor significantly higher, which may be based on Utrerana’s acknowledged darker yolk color when compared to laying lineages’ yolk color [4]. The higher pigmentation found in some strains may be due to different genetic capabilities to absorb and deposit pigments in yolk [30]. Different egg yolk color preferences have been reported between northern and southern European countries [31], with a taste towards intensely colored (golden-orange) yolks in southern countries, contrasting with what occurs in the majority of consumers worldwide, where consumers show a greater affinity towards brighter yolks. Similarly, consumers of ecologic/organic eggs generally accept paler yellow yolks, as reported by Grashorn [32].

Yolk color has been reported to depend directly on the carotenoid level and on the proportion between yellow and red carotenoids in the feed provided to laying hens. The content of yellow carotenoids (lutein, zeaxanthin, cryptoxanthin, violaxanthin, ethyl ester of β-apo-8′-carotenoic acid, β-apo-8′-carotenal) stabilize the yellow color in the yolk, but do not intensify it. Contrastingly, for a rather intense, golden-orange color, red carotenoids, such as capsanthin/capsorubin, canthaxanthin, and citranaxanthin have to be added to the feed. Red carotenoids cover yellow carotenoids, and if their content is further increased then the yolk color presents a pinkish or red tone [32].

A stronger egg odor has been attributed to dual-purpose hens when compared to laying hens [33]. Additionally, native breeds’ eggs have reported similar or superior values for aroma than improved breeds eggs [34]. These results contrast with those of Olugbemi et al. [35], who did not find significant differences between commercial laying hens and local breeds. Highly significant differences have been reported for egg composition across hen breeds and avian species, particularly regarding egg volatiles, fatty acids content, and albumen proteome composition [36]. Supportive findings by Rizzi and Marangon [34] indicated that dual-purpose hens present a stronger flavor. Haunshi et al. [34] also found flavor acceptability was significantly higher in local breeds than in improved breeds. No significant differences were found in either the texture across egg types, or in the literature for the texture of processed scrambled eggs belonging to hens fed on alternative products to dietary molt [37].

The Utrerana eggs’ overall score was significantly higher than commercial eggs’ score, agreeing with earlier reports [34]. The on-plate broken egg visual values were significantly lower in commercial eggs than ecologic and Utrerana eggs. This suggests that the Utrerana egg’s higher proportion of yolk and a darker yolk [4] would have a greater market acceptability. Panelists agreed very much on the whole egg visual value, as previously reported [4]. Furthermore, external appearance and eggshell color has been suggested to hold some positive correlations with egg quality parameters [38].

Panelists’ agreement was higher for fish consumption; drug prohibition; commercial, free range and ecological egg price; ecological product status relevance and egg in salad. Contextually, total drug banning led to the promotion of the consciousness of drug use, which develops a popular sense towards the effects of antibiotics and growth promoters on food safety and health in farms in Europe, which may be the basis for the panelists’ high agreement on the subject [39]. Furthermore, it can be inferred that when prices are higher, environmentally friendly food production and ecological products are regarded as secondary priorities [40]. Still, a tendency for consumers to pay higher prices for these environmentally-friendly products has been described in the literature [41].

Panelists agreed with respect to the environment, use of eggs in dessert and academic level (second dimension). Food consumption and production trends and patterns are one of the main causes of environmental pressure. Consumers are aware of this; hence, their consumption choices guide the search for more sustainable productive systems [42]. Not only did the PPB group consume more eggs, but also scored sensory attributes higher than the PPA group (non-usual eggs consumers, <4 eggs/week) and were more conscious about ecological production. Contrastingly, PPA reported a higher fruit, fish and dairy consumption, contrasting with the more frequent PPB meat and ecological products consumption habits. A lower frequency of consumption of eggs or meat in habitual fish consumers has been observed [43]. This could be explained by the division of panelists depending on their consumption livestock derived products. European diets are characterized by a high intake of livestock products (meat, dairy and eggs) [44].

The PPA group scored product closeness lower, was not acquainted with the Utrerana breed, misattributed a higher price to Utrerana eggs than to other egg types, scored ecological product status relevance lower and provided a relatively low importance to the product being linked to an Andalusian autochthonous and endangered breed. Contrastingly, the PPB group was acquainted with the Utrerana breed and scored the product and local endangered breeds-based production systems higher. Links with local breeds and their products starting from childhood improve their marketing strategies [45]. Hence, alternative markets may help conserving endangered native breeds and valuable animal genetic resources, as consumer demand for specialty livestock products and the willingness of consumers to pay for them largely depends on their lack of availability and their knowledge on the breed involved [46].

The functions of eggs, like coagulating, foaming, emulsifying and contributing nutrients, make them a useful ingredient in a lot of gastronomic preparations [47]. PPB provided a higher cuisine applicability to Utrerana eggs (desserts, appetisers, pasta, soup, salad and main courses) than PPA. Professionals acquainted with the product and its qualities, find a greater applicability than those who are not familiar. Polesel et al. [48] suggested egg consumption as an indicator of a diet rich in foods such as desserts and meat.

Women under 20 years old with no studies fit the PPA, while PPB would be characterized by men from 21 to 40 years old with secondary studies. These results agree with those of a previous study reporting that men usually consume more eggs than women, and individuals of 20–30 years old had the lowest odds of consuming eggs [49]. Stefanikova et al. [50] suggested men eat more meat, eggs and milk, while women eat more fruit and vegetables. Another study conducted in college students and nutrition educators suggested men consumed more meat, poultry, fish and eggs, while women consumed more vegetables and fruit [51]. Bejaei et al. [52] showed that free-run, free range, and organic eggs consumers have higher education levels compared with consumers of other egg types; this is supported by the fact that PPB–which valued more highly ecological and local hens’ eggs, than commercial ones–also presented higher education levels.

The differentiated quality of a product can be protected in markets offering a wider scope of valuable products, adapted to the consumer’s special needs [53]. Breed choice is mainly prescribed by the regulations of the producers, while the high quality of the products is appreciated by a small group of consumers, which indirectly promotes a local breed’s preservation [54]. Customer profile analysis adds value to the formulation of investment projects, providing information on consumers’ reactions to alternative products, generated through innovation or trend. In this context, the preferences of the target market among similar products, their purchase incentives, needs, among others, must be established. In this way, market research and the analysis of data collected enables the issuing of a diagnosis on the viability of the product in question, in turn translating into the sustainability of the breed that it originated from.

## 5. Conclusions

Involving autochthonous breeds, such as Utrerana, in common production systems and commercial chains seeking the characterization of differentiated products could be the key to improving profitability in future sustainable poultry productions. Defining different attitudes of costumers towards eggs may help outlining potential strategies for the design and implementation of marketing campaigns, indirectly identifying those sectors to which a greater effort should be made in an attempt to revalue a native breed’s egg products. These profiles may also suggest strategies on how to successfully achieve the aim of covering the currently increasing demand for non-conventional quality products linked to particular breeds and production systems from markets that are different from those normally established for classical highly productive systems.

## Figures and Tables

**Figure 1 animals-09-00920-f001:**
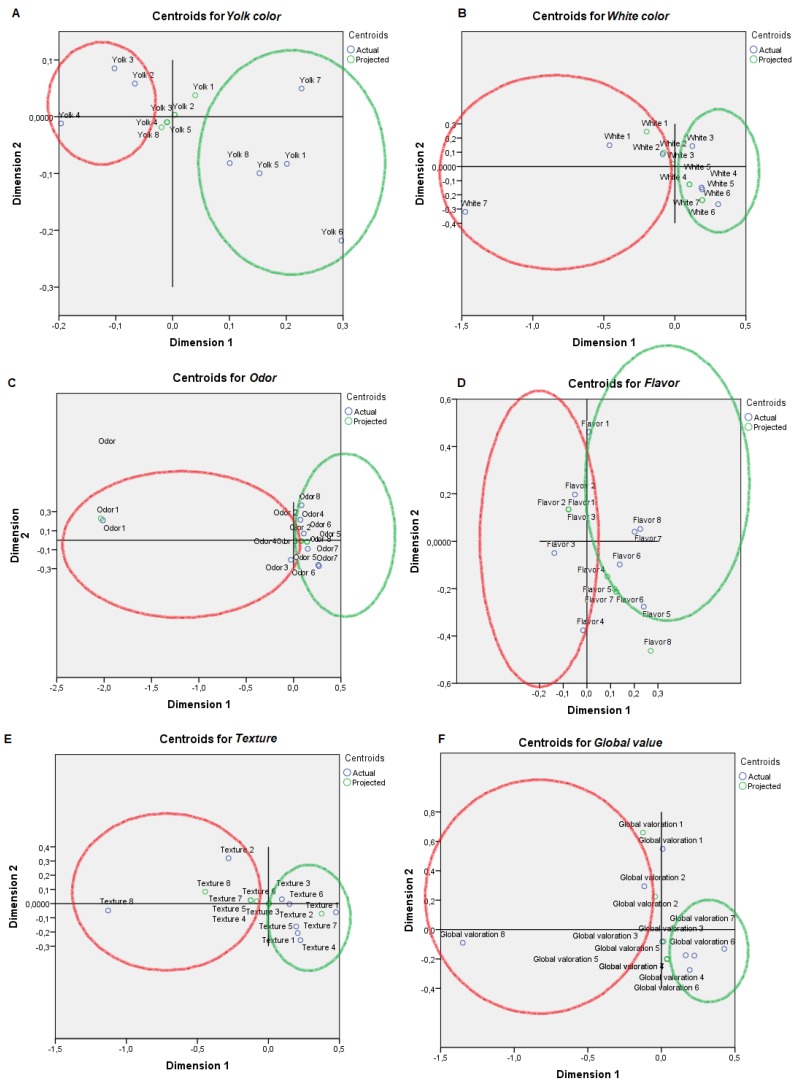
Object scores plot visualization of Professional Customer Profiles in regards egg sensory attributes, egg consumption non-affine profile or PPA (red), and affine profile or PPB (green). Egg sensory attributes were as follows: (**A**) Yolk color, (**B**) White color, (**C**) Odor, (**D**) Flavor, (**E**) Texture, (**F**) Global value, (**G**) Whole egg value and (**H**) On plate broken egg value.

**Figure 2 animals-09-00920-f002:**
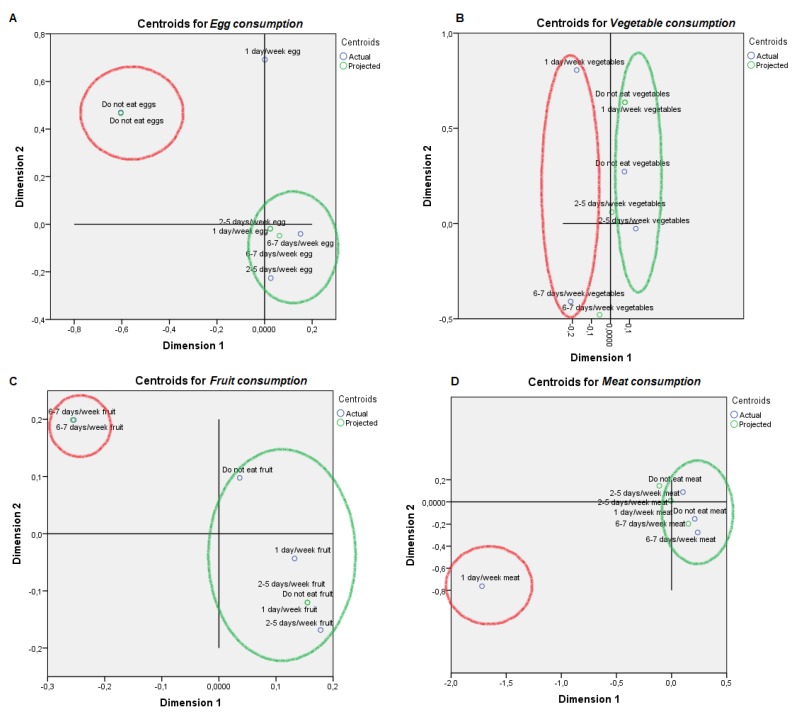
Object scores plot visualization of Professional Customer Profiles with regards panelist diet habits, egg consumption non-affine profile or PPA (red), and affine profile or PPB (green). Panelist diet habits were as follows: (**A**) Egg consumption, (**B**) Vegetable consumption, (**C**) Fruit consumption, (**D**) Meat consumption, (**E**) Fish consumption, (**F**) Dairy consumption, (**G**) Number of eggs per week and (**H**) Ecological consumer.

**Figure 3 animals-09-00920-f003:**
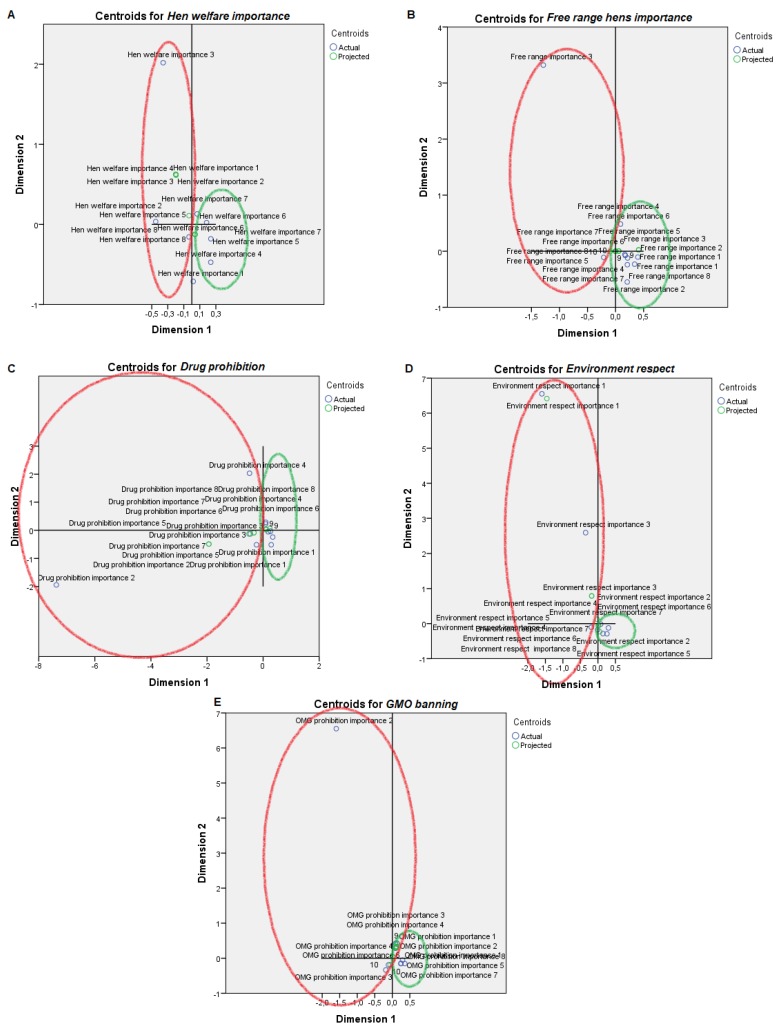
Object scores plot visualization of Professional Customer Profiles with regards production context awareness, egg consumption non-affine profile or PPA (red), and affine profile or PPB (green). Production context awareness was scored through: (**A**) Hen welfare importance, (**B**) Free range hens importance, (**C**) Drug prohibition, (**D**) Environment respect and (**E**) GMO banning.

**Figure 4 animals-09-00920-f004:**
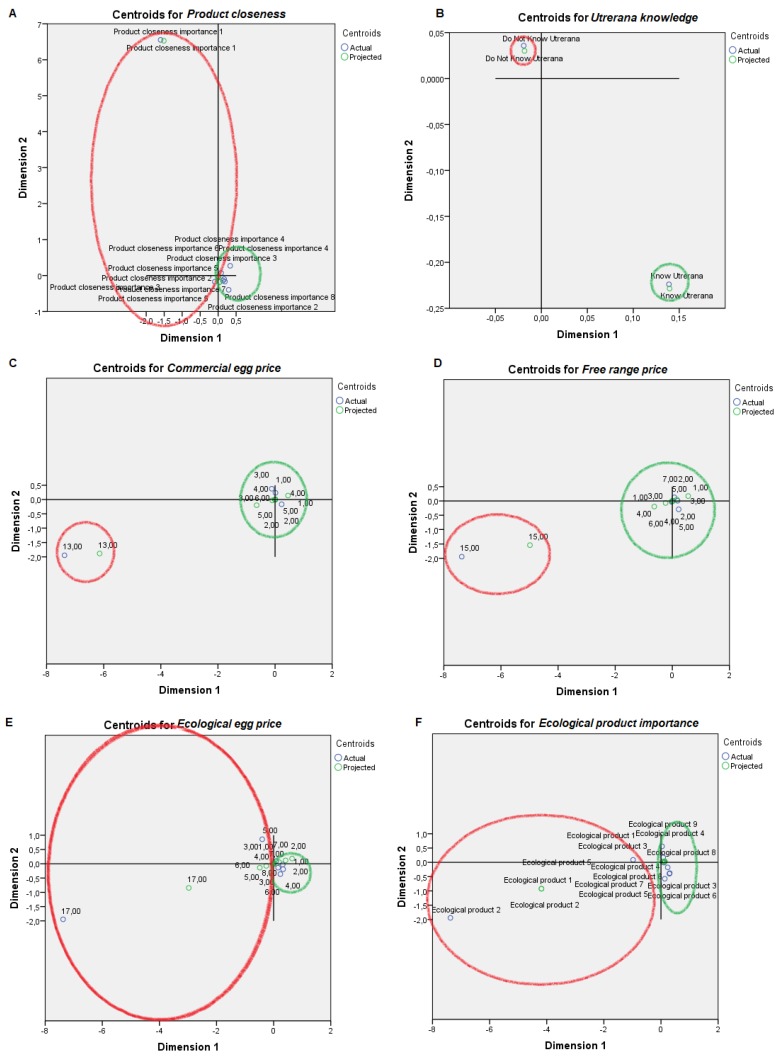
Object scores plot visualization of Professional Customer Profiles with regards product consciousness, egg consumption non-affine profile or PPA (red), and affine profile or PPB (green). Product consciousness was scored through: (**A**) Product closeness, (**B**) Utrerana knowledge, (**C**) Commercial egg price, (**D**) Free range price, (**E**) Ecological egg price, (**F**) Ecological product importance, (**G**) Product deriving from an Andalusian autochthonous breed, (**H**) Product deriving from an autochthonous endangered breed and (**I**) Product having a seasonal nature.

**Figure 5 animals-09-00920-f005:**
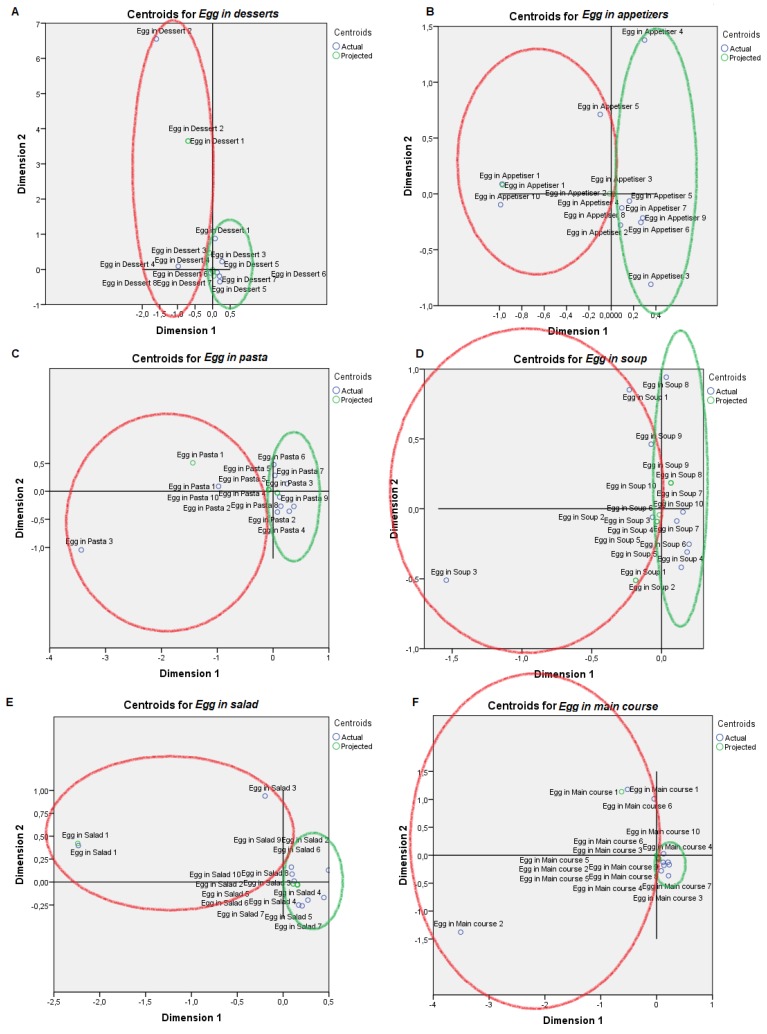
Object scores plot visualization of Professional Customer Profiles with regards cuisine applicability, egg consumption non-affine profile or PPA (red), and affine profile or PPB (green). Cuisine applicability was scored through: (**A**) Egg applicability in desserts, (**B**) in appetizers, (**C**) in pasta, (**D**) in soup, (**E**) in salad and (**F**) in main course.

**Figure 6 animals-09-00920-f006:**
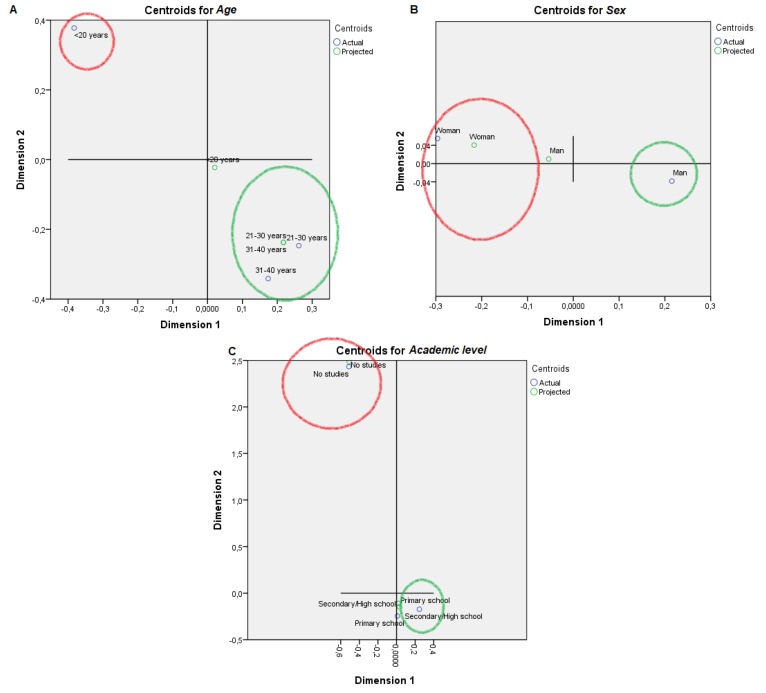
Object scores plot visualization of Professional Customer Profiles with regards professional characterization, egg consumption non-affine professional profile or PPA (red), and affine profile or PPB (green). Professional characterization was determined through: (**A**) Age, (**B**) Sex and (**C**) Academic level.

**Table 1 animals-09-00920-t001:** Cronbach’s Alpha, Cohen’s kappa Intra Class Correlation Coefficient and 95% confidence intervals for interobserver reliability testing and scale consistency sorted by egg type.

Egg Type			Cronbach’s Alpha				
Commercial			0.800				
Utrerana			0.826				
Ecologic			0.829				
Egg type	Measure type	Intraclass Correlation	95% Confidence Interval	F Test	df1	df2	Significance
Commercial	Single	0.105	0.071–0.158	5.003	63	2079	0.00
	Average	0.800	0.723–0.864	5.003	63	2079	0.00
Utrerana	Single	0.122	0.084–0.180	5.733	63	2079	0.00
	Average	0.826	0.758–0.882	5.733	63	2079	0.00
Ecologic	Single	0.125	0.086–0.183	5.843	63	2079	0.00
	Average	0.829	0.763–0.884	5.843	63	2079	0.00

**Table 2 animals-09-00920-t002:** Eigenvalues for the two-dimensional solution of nonlinear canonical correlation analysis for Utrerana native hen egg sensory attributes (yellow), panelist diet habits (red), production context awareness (purple), product consciousness (green), cuisine applicability (blue) and panelist characterization (grey) as perceived by cuisine instructed panelists (n = 192).

	Egg Sensory Attributes	Panelist Diet Habits	Production Context Awareness	Product Consciousness	Cuisine Applicability	Panelist Characterization	Mean	Eigenvalue
Dimension 1	0.446	0.069	0.46	0.055	0.081	0.849	0.327	0.673
Dimension 2	0.816	0.266	0.212	0.159	0.214	0.465	0.355	0.645
FIT	1.262	0.335	0.672	0.213	0.294	1.315	0.682	1.318

**Table 3 animals-09-00920-t003:** Model partitioning fit and loss analysis for Utrerana native hen egg sensory attributes (yellow), panelist diet habits (red), production context awareness (purple), product consciousness (green), cuisine applicability (blue) and panelist characterization (grey) as perceived by cuisine instructed panelists (n = 192).

Set.	Variables	Categories	Multiple Fit			Single Fit			Single Loss		
			Dimension 1	Dimension 2	Sum	Dimension 1	Dimension 2	Sum	Dimension 1	Dimension 2	Sum
Egg sensory attributes	Yolk color	8	0.060	0.015	0.074	0.052	0.013	0.065	0.007	0.002	0.009
	White color	7	0.078	0.026	0.105	0.077	0.019	0.096	0.002	0.007	0.008
	Odor	8	0.088	0.022	0.110	0.085	0.005	0.090	0.003	0.017	0.021
	Flavor	8	0.048	0.030	0.078	0.039	0.024	0.062	0.009	0.006	0.015
	Texture	8	0.045	0.095	0.140	0.032	0.093	0.126	0.013	0.002	0.015
	Overall value	8	0.009	0.099	0.107	0.005	0.097	0.103	0.004	0.001	0.005
	Whole egg visual value	8	0.311	0.068	0.379	0.302	0.047	0.349	0.010	0.021	0.030
	On plate broken egg visual value	8	0.017	0.042	0.059	0.012	0.041	0.053	0.005	0.002	0.006
Panelist diet habits	Egg consumption	4	0.002	0.162	0.164	0.002	0.161	0.163	0.000	0.001	0.001
	Vegetable consumption	4	0.032	0.254	0.286	0.032	0.249	0.281	0.000	0.005	0.005
	Fruit consumption	4	0.014	0.134	0.148	0.013	0.132	0.145	0.001	0.002	0.003
	Meat consumption	4	0.005	0.017	0.022	0.005	0.016	0.021	0.001	0.001	0.001
	Fish consumption	4	0.906	0.067	0.974	0.906	0.055	0.961	0.001	0.012	0.013
	Dairy consumption	4	0.006	0.116	0.122	0.004	0.101	0.105	0.002	0.015	0.017
	Number of eggs per week	5	0.022	0.525	0.547	0.021	0.524	0.545	0.001	0.001	0.002
	Ecological consumer	2	0.000	0.006	0.007	0.000	0.006	0.007	0.000	0.000	0.000
Production context awareness	Hen welfare	8	0.040	0.025	0.065	0.003	0.008	0.011	0.038	0.017	0.054
	Free range hens	10	0.230	0.026	0.256	0.216	0.001	0.217	0.014	0.025	0.039
	Drug prohibition	10	0.985	0.108	1.093	0.644	0.073	0.717	0.341	0.035	0.376
	Environment respect	8	0.069	0.854	0.923	0.036	0.798	0.834	0.033	0.056	0.089
	GMO banning	10	0.048	0.024	0.072	0.045	0.012	0.058	0.002	0.012	0.014
Product consciousness	Product closeness	8	0.055	0.676	0.732	0.055	0.674	0.729	0.001	0.003	0.003
	Utrerana knowledge	2	0.000	0.002	0.002	0.000	0.002	0.002	0.000	0.000	0.000
	Commercial egg price	13	0.021	0.051	0.072	0.020	0.050	0.071	0.001	0.001	0.001
	Free range egg price	15	0.002	0.051	0.052	0.001	0.051	0.052	0.000	0.000	0.001
	Ecological egg price	17	0.002	0.025	0.027	0.000	0.001	0.001	0.002	0.024	0.026
	Ecological product	9	2.097	0.080	2.177	2.094	0.056	2.150	0.002	0.024	0.027
	Andalusian autochthonous breed product	9	0.803	0.009	0.812	0.803	0.001	0.804	0.000	0.008	0.009
	Endangered breed product	10	0.011	0.140	0.150	0.008	0.127	0.135	0.003	0.013	0.016
	Seasonal product	10	0.004	0.019	0.023	0.001	0.001	0.002	0.004	0.018	0.021
Cuisine applicability	Desserts	8	0.039	0.641	0.679	0.023	0.526	0.548	0.016	0.115	0.131
	Appetizers	10	0.017	0.340	0.357	0.002	0.313	0.315	0.015	0.027	0.042
	Pasta	10	0.032	0.222	0.253	0.021	0.213	0.234	0.010	0.009	0.019
	Soup	10	0.011	0.247	0.258	0.006	0.242	0.248	0.006	0.005	0.011
	Salad	10	3.594	0.255	3.849	3.585	0.215	3.800	0.009	0.040	0.049
	Main course	10	2.323	0.924	3.247	2.319	0.904	3.223	0.004	0.020	0.024
Panelist characterization	Age	3	0.193	0.046	0.239	0.192	0.046	0.239	0.000	0.000	0.000
	Sex	2	0.183	0.012	0.195	0.183	0.012	0.195	0.000	0.000	0.000
	Academic level	3	0.015	0.475	0.490	0.008	0.475	0.483	0.007	0.000	0.007

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
