# Peer review of "Sensory Preference and Professional Profile Affinity Definition of Endangered Native Breed Eggs Compared to Commercial Laying Lineages’ Eggs"

_animals, 2019, doi:10.3390/ani9110920_

Round 1
Reviewer 1 Report
The aim of this study was to aimed to determine the ability of panelists to discriminate organoleptic characteristics across egg type categories basing on hedonic scales and determination of differences in the studied sensory characteristics of eggs from organic and traditional farming (commercial hybrids of White Leghorn and native Spanish Utrerana hen ). The results of this experiment have cognitive elements for hen eggs consumers and the general public. The number of panelists assessing the examined features was sufficient. Preparation panels for assessment and description of research methods are correct. A rich statistical study deserves attention. The discussion is well carried out and exhausting. References well chosen. Before publishing in Animals, the paper requires additions and corrections. The list of proposed changes is given below:
L57: please add information on the colour of the Utreran hen eggshell and hen age - egg weight was 64 g, for the entire laying period or at the specified age?
L36 Please provide the commercial name of the White Leghorn laying hen set
L60 please provide the world egg production volume (million tonnes) in 2017 according to FAOSTAT
L109, the place of residence (village, town), material status of panelists could also have been evaluated
L113 + please provide information on the component (recipe) and chemical composition of the compound feed used in feeding hen, whether it was the same in the production of traditional (commercial hybrids and native hen flock) and organic eggs. The composition of the diet has an impact on the chemical composition of eggs, fatty acid profile, AA, content of minerals, xanthophyll pigments and carotene in eggs that affect the sensory properties studied. Please provide information on the maintenance system of commercial hens, native bred and hens from which organic eggs have been obtained
L140 + no description of the assessment scales used for the examined sensory characteristics of eggs: yolk color, white color, smell, flavor, texture, overall value
L310 and the rest paper color instead of color
L363 please provide information on differences in consumer preferences in different European countries regarding the color of the yolk, describe the effect of diet composition on the color of the yolk (content of xanthophyll pigments and carotene)
L381 Please post a discussion on the availability of calcium by layers as they grow older, the optimal ratio of calcium to phosphorus in the diet of laying hens, the impact of room temperature on shell thickness and susceptibility to breakage
L389: antibiotics, but what? feed chickens have not been used and feed antibiotics are not used.
L491: add search terms in the FAOSTAT database
Author Response
All the team responsible for this paper acknowledge the comments from the reviewers and editor, as they help to improve the quality of our manuscript. In the following paragraphs, we will describe and address how referees’ new recommendations were followed. A point-by-point response to comments is provided as well as a file where changes are highlighted.
Comments and Suggestions for Authors
The aim of this study was to aimed to determine the ability of panelists to discriminate organoleptic characteristics across egg type categories basing on hedonic scales and determination of differences in the studied sensory characteristics of eggs from organic and traditional farming (commercial hybrids of White Leghorn and native Spanish Utrerana hen ). The results of this experiment have cognitive elements for hen eggs consumers and the general public. The number of panelists assessing the examined features was sufficient. Preparation panels for assessment and description of research methods are correct. A rich statistical study deserves attention. The discussion is well carried out and exhausting. References well chosen. Before publishing in Animals, the paper requires additions and corrections. The list of proposed changes is given below:
L57: please add information on the colour of the Utreran hen eggshell and hen age - egg weight was 64 g, for the entire laying period or at the specified age?
Response: Information required by reviewer was enclosed.
L36 Please provide the commercial name of the White Leghorn laying hen set
Response: White Leghorn laying hen set belonged to Leghorn Lohmann LSL-Classic as it has been included in the abstract and body text as suggested by reviewer.
L60 please provide the world egg production volume (million tonnes) in 2017 according to FAOSTAT
Response: The information requested by the reviewer was provided as suggested.
L109, the place of residence (village, town), material status of panelists could also have been evaluated
Response: We agree with the reviewer on the fact that having added place of residence and material status to the study may have been a good option. However, it was not possible as considering such factors may have also implied subdividing the sample in several categories, which considering the difficulty to gather a panellist sample size already limited but enough to perform the study, could have made the conclusions drawn biased or invalid.
L113 + please provide information on the component (recipe) and chemical composition of the compound feed used in feeding hen, whether it was the same in the production of traditional (commercial hybrids and native hen flock) and organic eggs. The composition of the diet has an impact on the chemical composition of eggs, fatty acid profile, AA, content of minerals, xanthophyll pigments and carotene in eggs that affect the sensory properties studied. Please provide information on the maintenance system of commercial hens, native bred and hens from which organic eggs have been obtained.
Response: Further details regarding the diet and its composition and husbandry practices and management system has been added following the suggestion by the reviewer.
L140 + no description of the assessment scales used for the examined sensory characteristics of eggs: yolk color, white color, smell, flavor, texture, overall value
Response: We included the description of the assessment scales used for the evaluation of sensory characteristics as suggested by reviewer.
L310 and the rest paper color instead of color
Response: We do not know if we have understand what the reviewer suggestion was. However, we understood that the reviewer suggestion was for us to change colour to color across the whole manuscript.
L363 please provide information on differences in consumer preferences in different European countries regarding the color of the yolk, describe the effect of diet composition on the color of the yolk (content of xanthophyll pigments and carotene).
Response: We added two parapgraphs providing the information requested by the reviewer, regarding European consumer taste towards yolk color and color relationship with yellow and red carotenes.
L381 Please post a discussion on the availability of calcium by layers as they grow older, the optimal ratio of calcium to phosphorus in the diet of laying hens, the impact of room temperature on shell thickness and susceptibility to breakage
Response: We appreciate the reviewer comment but we feel there was a misunderstanding regarding the concept of broken egg visual value, as with broken egg visual value we referred to the visual score provided to the egg once broken an presented on a plate. For this reason, we clarified this concept and added on plate before broken egg through the manuscript to avoid potential misunderstandings. Bearing this in mind, the discussion requested does not fit the information provided in the paragraph, and whole increase text length considerably.
L389: antibiotics, but what? feed chickens have not been used and feed antibiotics are not used.
Response: By drug use awareness, we refer to the attitude of panelists towards this topic, not to the fact that any drug was used in the study. However, we understand the concern of the reviewer and clarified the information in the text in this regard.
L491: add search terms in the FAOSTAT database
Response: Search terms were added as requested by the reviewer.
To this end, we thank you for your time and attention and for considering this manuscript.
Reviewer 2 Report
The study conducted by Ariza et al. was mainly compared Utrerana native hen eggs’ sensory properties to laying lineage’s commercial and ecological ones through free-choice profiling.The topic is interesting. There are fewer minor comments.
The Simple Summary looks a litter bit long. Could you please reduce it. Adding the 'In conclusion' in a proper place of the Abstract will be helpful. There are many paragraphs in the introductions. Some of them only have 3-4 rows. The authors might need to combined some of them. In the results, all the figures is not clear showed in the review version. All the figures can not be read clearly. Please upload the high resolution pictures. There are many paragraphs in the discussion section. Some of them only have 3-4 rows. The authors might need to combined some of them.Author Response
All the team responsible for this paper acknowledge the comments from the reviewers and editor, as they help to improve the quality of our manuscript. In the following paragraphs, we will describe and address how referees’ new recommendations were followed. A point-by-point response to comments is provided as well as a file where changes are highlighted.
Comments and Suggestions for Authors
The study conducted by Ariza et al. was mainly compared Utrerana native hen eggs’ sensory properties to laying lineage’s commercial and ecological ones through free-choice profiling.The topic is interesting. There are fewer minor comments.
The Simple Summary looks a litter bit long. Could you please reduce it.
Response: We reduced the simple summary from 200 words to 162 as suggested by the reviewer.
Adding the 'In conclusion' in a proper place of the Abstract will be helpful.
Response: We added the connector “In conclusion”, to the line in the abstract in which we fell it would be more appropriate.
There are many paragraphs in the introductions. Some of them only have 3-4 rows. The authors might need to combined some of them.
Response: Paragraphs in the introduction were combined in order to prevent multiple segmentation and improve readability as suggested by the reviewer.
In the results, all the figures is not clear showed in the review version. All the figures can not be read clearly. Please upload the high resolution pictures.
Response: We uploaded the resolution of the images as much as we possibly could. Furthermore, when images were not displayed properly, we splitted them so as to improve readability.
There are many paragraphs in the discussion section. Some of them only have 3-4 rows. The authors might need to combined some of them.
Response: We agree with the reviewer’s suggestion and combined paragraphs comprising 3 or 4 sentences into larger ones as requested.
To this end, we thank you for your time and attention and for considering this manuscript.